# A Rare Case of Radiation-Induced Liver Disease in Treated Abdominal Lymphoma Showing High [^18^F]FDG Avidity and Low EOB Uptake Proportional to the Irradiation Dose

**DOI:** 10.3390/diagnostics12020504

**Published:** 2022-02-16

**Authors:** Aya Usami, Kota Yokoyama, Junichi Tsuchiya, Yoshihiro Umezawa, Kazuma Toda, Ukihide Tateishi, Ryoichi Yoshimura

**Affiliations:** 1Department of Radiation Therapeutics and Oncology, Tokyo Medical and Dental University, Tokyo 113-8510, Japan; aya_on_no@yahoo.co.jp (A.U.); goburei48000@mtc.biglobe.ne.jp (K.T.); ysmrmrad@tmd.ac.jp (R.Y.); 2Department of Diagnostic Radiology and Nuclear Medicine, Tokyo Medical and Dental University, Tokyo 113-8510, Japan; tuwu11@gmail.com (J.T.); ttisdrnm@tmd.ac.jp (U.T.); 3Department of Hematology, Tokyo Medical and Dental University, Tokyo 113-8510, Japan; ume.hema@tmd.ac.jp

**Keywords:** radiation-induced liver disease, FDG PET/CT, EOB-MRI, DLBCL

## Abstract

A 44-year-old woman presented with high [^18^F]FDG uptake liver lesion after six courses of R-CHOP and radiotherapy for abdominal DLBCL, which was misdiagnosed as a hepatic invasion. EOB–MRI showed slight T2 hyperintensity, low-intensity DWI, and decreased EOB uptake in the hepatocellular phase. Compared with the pretreatment planning CT, the liver lesion coincided with the area of >40.5 Gy, resulting in the diagnosis of RILD. At the follow-up [^18^F]FDG PET/CT 7 months after irradiation, the abnormal liver uptake disappeared. Comparing [^18^F]FDG PET/CT, EOB–MRI, and planning CT can lead to the correct diagnosis of RILD and avoid unnecessary biopsies and treatment changes.

**Figure 1 diagnostics-12-00504-f001:**
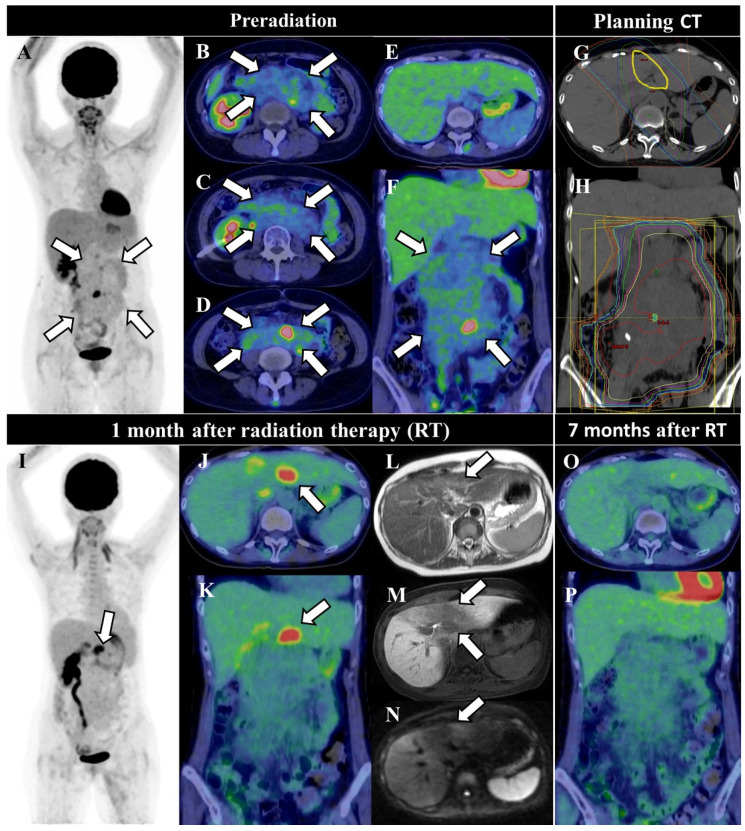
A 44-year-old woman with abdominal diffuse large B-cell lymphoma (DLBCL) received six courses of rituximab, cyclophosphamide, doxorubicin, vincristine, and prednisone. Treatment assessment 2-Deoxy-2-[^18^F]fluoroglucose ([^18^F]FDG) positron emission tomography/computed tomography (PET/CT) showed high [^18^F]FDG uptake scattered over a large mass in the abdomen (**A**–**F** arrows). She received additional radiotherapy (RT) with 45 Gy in 25 fractions with three-dimensional conformal therapy (**G**,**H**). A second PET/CT performed for treatment assessment 1 month after irradiation showed a decrease in [^18^F]FDG uptake in the abdominal mass, but a new focal accumulation was observed in the left lobe of the liver, with a standardized maximum uptake of 7.0, suggesting hepatic infiltration of the DLBCL (**I**–**K**). Ethoxibenzyl (EOB)–magnetic resonance imaging (MRI) showed slight T2 hyperintensity (**L**) and decreased EOB uptake in the hepatocellular phase (**M**). Diffusion-weighted imaging (DWI) showed low intensity (**N**). Compared with the pretreatment planning CT images, it can be recognized that the liver lesion coincides with the area of 90% of the prescribed dose, i.e., the area of >40.5 Gy (**G** yellow line), and that the highest dose area has the highest [^18^F]FDG avidity (**I**–**K** arrows) and lowest EOB uptake (**M** arrow). Seven months after irradiation, [^18^F]FDG uptake in the lesion had decreased (**O**,**P**), which was consistent with radiation-induced liver disease (RILD). No elevations in liver enzymes or soluble interleukin-2 receptors were observed during the study period. RILD is usually asymptomatic and not associated with liver enzyme elevations but is known to have a high [^18^F]FDG uptake [1], occurring in 3–8% of patients with esophageal cancer treated with chemoradiation who are reevaluated by PET/CT [2]. Only a few reports have reported the use of [^18^F]FDG PET/CT for RILD other than esophageal cancer [3]. RILD usually occurs 4–8 weeks at the end of RT [4], but it has been reported to occur as early as 2 weeks and as late as 7 months after RT [5]. The disease has been reported to generally disappear during follow-up, and a report revealed that it disappeared as early as 5 months. In the case of ML, this condition should be understood because it is just around the time when the follow-up PET/CT is performed. In CT, RILD shows low attenuation in the area irradiated over 45 Gy [6], whereas [^18^F]FDG PET/CT shows high uptake at ≥40 Gy [2]. In MRI, T2 hyperintensity and low EOB uptake are also reported [7]. The mechanism of increased [^18^F]FDG uptake is thought to be active inflammation [8], which can be broadly called “pseudoprogression” [9]. Interestingly, in this case, DWI showed low signal, which is unusual for acute inflammation, and could be explained by some fibrosis due to radiation, which may be useful in differentiating liver invasion. To the best of our knowledge, this is the first case of RILD incidentally detected by follow-up PET/CT after RT for abdominal ML. Since the mechanism and clinical significance of [^18^F]FDG uptake in RILD are not clear [2,9], it is unclear whether it should be followed up. Pathologically, it has been reported to be necrosis or veno-occlusive disease [10,11], but most of them are in patients with hepatocellular carcinoma with cirrhosis or in patients with clinical liver failure, which is different from asymptomatic cases detected by [^18^F]FDG PET/CT without elevated liver enzymes [1,2,9]. In addition, a patient with high [^18^F]FDG uptake in the focal liver lesion has been reported to have normal hepatocytes on biopsy [2]. It has been reported that RILD is more likely to be severe under conditions such as liver cirrhosis, high dose, and whole-liver radiation [4,12,13], and careful follow-up may be necessary under these conditions. In studies of lower esophageal cancer, RILD was diagnosed by comparing the PET findings with the irradiated area, and biopsy was avoided in most cases [1,2,6,9,14]. If high [^18^F]FDG uptake is observed in the liver 2 weeks to 7 months after radiotherapy, comparing [^18^F]FDG PET/CT, EOB-MRI, and planning CT is important to correctly diagnose RILD and to avoid unnecessary biopsies and treatment changes.

## Data Availability

Not applicable.

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
