# Peer review of "A Rare Case of Radiation-Induced Liver Disease in Treated Abdominal Lymphoma Showing High [18F]FDG Avidity and Low EOB Uptake Proportional to the Irradiation Dose"

_diagnostics, 2022, doi:10.3390/diagnostics12020504_

Round 1

Reviewer 1 Report

This case approach an interesting and practical issue in the diagnostic follow-up of disease by FDG-PET/CT. Metastases and radiation-induced liver disease may manifest as increased foci of FDG avidity at PET scanning. To be sure, an histological confirmation is the best way to obtain a differential diagnosis, but the evolution and appearance of the cases of radiation hepatitis and metastatic disease seem sufficiently distinctive.

The close comparison of PET ann MRI evolutive finding may be interesting to take into account this challenge situation as the authors had ilustrated in this case.

A couple of questions for the authors: can be use the ‘pseudoprogression’ concept in the discussion setting? Doeas the authors think a hepatitis may be the potential cause of their finding?

Reviewer 2 Report

The case presented can be considered for publication. However, the discussion of the case is not very thorough and would benefit from a better framing of the case in terms of clinical presentation and above all a discussion that would take into consideration the current state of the post-therapy follow-up procedures and the possible implications suggested by the case presented.
It is suggested to present the images with the same color scales

Round 2

Reviewer 2 Report

The paper has been significantly improved
It can be published